# Screening and Analysis of Possible Drugs Binding to PDGFRα: A Molecular Modeling Study

**DOI:** 10.3390/ijms24119623

**Published:** 2023-06-01

**Authors:** Matteo Mozzicafreddo, Devis Benfaremo, Chiara Paolini, Silvia Agarbati, Silvia Svegliati Baroni, Gianluca Moroncini

**Affiliations:** 1Department of Clinical and Molecular Sciences, Marche Polytechnic University, 60126 Ancona, Italy; 2Clinica Medica, Department of Internal Medicine, Marche University Hospital, 60126 Ancona, Italy

**Keywords:** PDGF receptor (PDGFR), systemic sclerosis (SSc), molecular docking, molecular dynamics (MD), quantitative structure–activity relationship (QSAR), structure-based drug design (SBDD)

## Abstract

The platelet-derived growth factor receptor (PDGFR) is a membrane tyrosine kinase receptor involved in several metabolic pathways, not only physiological but also pathological, as in tumor progression, immune-mediated diseases, and viral diseases. Considering this macromolecule as a druggable target for modulation/inhibition of these conditions, the aim of this work was to find new ligands or new information to design novel effective drugs. We performed an initial interaction screening with the human intracellular PDGFRα of about 7200 drugs and natural compounds contained in 5 independent databases/libraries implemented in the MTiOpenScreen web server. After the selection of 27 compounds, a structural analysis of the obtained complexes was performed. Three-dimensional quantitative structure–activity relationship (3D-QSAR) and absorption, distribution, metabolism, excretion, and toxicity (ADMET) analyses were also performed to understand the physicochemical properties of identified compounds to increase affinity and selectivity for PDGFRα. Among these 27 compounds, the drugs Bafetinib, Radotinib, Flumatinib, and Imatinib showed higher affinity for this tyrosine kinase receptor, lying in the nanomolar order, while the natural products included in this group, such as curcumin, luteolin, and epigallocatechin gallate (EGCG), showed sub-micromolar affinities. Although experimental studies are mandatory to fully understand the mechanisms behind PDGFRα inhibitors, the structural information obtained through this study could provide useful insight into the future development of more effective and targeted treatments for PDGFRα-related diseases, such as cancer and fibrosis.

## 1. Introduction

Platelet-derived growth factor receptors alpha and beta (PDGFRα and PDGFRβ) are membrane receptors that play a key role in a variety of diseases, such as cancer [1,2,3,4,5], immune-mediated pathologies, such as systemic sclerosis (SSc) [6,7,8,9,10], and viral infection [11], by exerting function on the transduction of extracellular signals into the cell. PDGFRs are transmembrane glycoprotein dimer molecules consisting of an extracellular ligand-binding region divided into five Ig-like domains bound to an intracellular receptor tyrosine kinase (RTK) domain through a single transmembrane alpha helix. The tyrosine kinase domain is a typical type III RTK formed by a bilobal kinase, a juxtamembrane (JM) domain, and an activation loop (A-loop) [11,12]. The JM domain is inserted into the active site, while the A-loop (that starts with a conserved amino acid sequence DFG) controls access to it and partially blocks the ATP and substrate site in the inactive state of the enzyme. In the active state of the RTK, the DFG motif is located in a hydrophobic pocket close to the ATP binding site. When an inhibitor binds this pocket, the ATP binding site undergoes a conformational change that reversibly or irreversibly prevents its normal function [13].

Considering that the human PDGFRα has become one of the most important therapeutic targets for the above-mentioned diseases [4,5,6,7,8,14,15,16,17], its modulation/inhibition is a critical aspect that determines the investigation of new ligands or new information to design new effective drugs. As previously reported [18,19,20,21,22,23], inhibiting PDGFRα (overexpressed and/or abnormally activated) can reduce many types of cancer, including prostate, ovarian, breast, pancreatic, and liver cancers, by inactivating downstream signaling pathways that regulate cell proliferation, migration, and angiogenesis [24,25,26]. On the other hand, these signal pathways can induce proliferation and chemotaxis of myofibroblasts, collagen production, and adhesion in endothelial cells [27]. In pathological states, which are also correlated to viral infections, they are upregulated in patients with a progressive fibrotic disease such as SSc [28,29,30].

For these reasons, the structure-based drug design (SBDD), which promotes the development of novel drugs with a potential affinity for therapeutic targets, has turned out to be an essential tool for rapid and cost-efficient lead discovery. SBDD is a growing, iterative, and powerful approach that involves the structural evaluation of targets in the drug discovery process. It has the ability to reduce the time and cost of developing new drug lead molecules with potential therapeutic effects [31]. A virtual ligands database screening, followed by molecular docking, molecular dynamics (MD), and three-dimensional quantitative structure–activity relationship (3D-QSAR) analysis, as previously reported [2,3,5,32], provides an excellent workflow (Figure 1) to assess an efficient SBDD.

Here, we analyzed 27 molecules selected from the results of structure-based virtual screening following this workflow and characterizing the most important features of an ideal ligand/inhibitor for the intracellular RTK domain of human PDGFRα.

## 2. Results

### 2.1. Virtual Screening of the Small Compounds Database

The structure-based virtual screening of the databases (as descripted in the Methods section) against the human intracellular PDGFRα (hiPDGFRα) provided an early version of the best compounds list (Appendix A). This was evaluated by taking the best ligands of each of the 5 drug-like chemical libraries, resulting in the 27 ligands reported in Table 1. Among these libraries, the Drugs-lib had the most involvement, showing ligands with higher affinities, although the natural products selected showed *K_d_* values in the sub-micromolar range, indicating medium–high affinities.

### 2.2. Molecular Docking

The molecular docking analysis, which was used to check the virtual screening results, provided affinity values reported in Table 1 and binding geometries of the ligand/hiPDGFRα complexes. These values are in the nanomolar/sub-micromolar range and confirm the ranking obtained by the structure-based virtual screening (Appendix A).

To validate the docking process, we compared the Imatinib/hiPDGFRα predicted complex with the available crystal structure, obtaining a ligand–ligand superimpose root mean square deviation (RMSD) value of 0.198 Å. This value is significantly lower than 1 Å, indicating a high reliability of the method.

All complexes involved the hiPDGFRα active site, although the docking box was set for the entire protein, indicating no other binding site. The four best ligands interacted with hiPDGFRα through the formation of five hydrogen bonds with Glu644, Thr674, Cys677, and Asp836, showing very similar geometries. As reported in Figure 2A, Bafetinib also interacted with the protein, forming six hydrophobic interactions with Val607, Glu644, Ile647, Thr674, Leu825, and Phe837, and one π-cation interaction with Lys627. This π-cation interaction was also formed in the Flumatinib/hiPDGFRα complex, which additionally showed nine hydrophobic interactions (with Leu599, Val607, Lys627, Glu644, Val658, Thr674, Leu825, Asp836, and Phe837) and a halogen bond between Glu644 and the trifluoromethyl group of the ligand (Figure 2B). This π-cation interaction, the halogen bond, and the distances and number of hydrophobic interactions between the ligand and the hiPDGFRα seem to be crucial for binding affinity.

Although the Imatinib/hiPDGFRα crystal structure shows seven hydrogen bonds (with Glu644, Met648, Thr674, Cys677, His815, Val816), a salt bridge (Asp836), two π-cation interactions (with His816 and Tyr676), and seven hydrophobic interactions (with Leu599, Lys627, Glu644, Val658, Ile672, Thr674, and Asp836), the predicted Imatinib/hiPDGFRα complex, after the molecular dynamics analysis, shows only five hydrogen bonds previously described and nine hydrophobic interactions (with Leu599, Val607, Lys627, Ile672, Thr674, Tyr676, Leu825, Asp836, and Phe837). This could be explained by the difference in temperature between the two types of tests (293 K for the crystallographic analysis and 300 K for the molecular dynamics). 

Regarding the binding between natural ligands such as curcumin and quercetin and hiPDGFRα, this involves only a minor number of hydrogen bonds and hydrophobic interactions, even if their strength and number establish a medium–high affinity (*K_d,pred_* = 0.2 μM and *K_d,pred_* = 0.7 μM, respectively). This still determines an excellent competition against ATP that has an affinity for the wildtype enzyme in the order of sub-millimolar (*K_m_* = 179.6 μM) [12], underscoring the potency of their enzymatic inhibition capabilities.

### 2.3. Molecular Dynamics

The stability of the best 4 predicted complexes, obtained through molecular docking, was analyzed using molecular dynamics simulation for 50 ns. The first four nanoseconds of these simulations are represented in Figure 3 since no further RMSD variations were established. The calculated RMSD of each ligand and complex trajectory with respect to the backbone indicates that the complexes achieve sufficient stability (RMSD < 0.17 nm, 1.7 Å) within less than 4 ns, except for Flumatinib (Figure 3G), which shows RMSD values of about 0.22 nm.

The RMSD trend of Bafetinib (Figure 3A) and Flumatinib (Figure 3G) with respect to the other ligands seems to require a few more picoseconds and nanoseconds, respectively, to reach final stability. This behavior could be explained by the establishment of the π-cation interaction.

The average short-range Coulombic and Lennard–Jones interaction energies calculated during the molecular dynamics analysis are reported in Table 2. As noted in the molecular docking section, the electrostatic contributions between Bafetinib and Flumatinib are essentially equivalent, while there is a significant difference in terms of intermolecular pair Lennard–Jones potential, indicating a more efficient binding of Bafetinib to the active site. 

### 2.4. Three-Dimensional Quantitative Structure–Activity Relationship (3D-QSAR) and ADMET Analyses

The three-dimensional QSAR analysis produced comparative molecular fields analysis (CoMFA) models considering the CoMFA potential or field, the best of which are summarized in Table 3 based on optimal principal components (OPC). Among these, MW, nHA, nHD, LogP, TPSA, rotatable bonds (RBs), molar refractivity (MR), length, and max length were taken into account and modulated to achieve the best model. Some of them are reported in Table 4 and Table 5.

Figure 4 shows the relationships between experimental and calculated activities by CoMFA models. The fitting statistical results, also reported in Table 3, indicate that the steric model (STE) is the more consistent and robust one, even if the electrostatic one (ELE) shows a relatively good squared correlation coefficient *r*^2^ and internal predictive coefficient *q*^2^. This would confirm the result of interaction energies obtained by Gromacs, which describes the best complex in terms of affinity as the one with the higher Lennard–Jones energy contribution, the forces involved in the steric effect.

ADMET predictions, in particular, some properties and relative descriptors of the 27 selected molecules, are reported in Table 4 and Table 5. A major part of these compounds satisfies the conditions regarding the acceptance of the first two indexes (Lipinski and Pfizer) related to absorption, permeability, and toxicity. Only Ditercalinium does not satisfy the conditions of these indexes since it presents the heaviest and longest molecule and the highest MR and LogP, which are out of range. Several compounds could cause skin sensitization and respiratory toxicity—although no compound would cause acute toxicity during oral administration.

## 3. Discussion

The PDGFRα is a critical factor that plays an essential role in regulating cell proliferation, survival, and chemotaxis [24,25,26,27]. Several studies have identified PDGFR as a potential target to treat pathologies, such as cancer and progressive fibrotic diseases [4,5,6,7,8,14,15,16,17], by searching for synthetic or natural compounds acting as ligands/inhibitors. Currently, however, no one has considered involving such large compound libraries for this purpose, such as those implemented in the MTiOpenScreen web server.

The details of binding interactions between human intracellular PDGFRα and the compounds selected after the structure-based virtual screening were revealed by molecular docking and the subsequent MD simulations. Molecular docking analysis, in addition to complexes’ affinities, was able to provide important critical issues regarding the geometries. The predicted affinities for the 27 selected complexes vary from the nanomolar order (1.01 nM for the Bafetinib/hiPDGFRα complex) to the micromolar one, a typical range of values for a good drug inhibition and reliable QSAR analysis. Some of the most stable complexes show a π-cation possible interaction with Lys627, halogen bonds with Glu644, and numerous hydrophobic interactions. 

MD analysis reveals that the complexes achieve their stability in a very short time and remain stable for all the MD simulations, demonstrating no substantial unfolding of the structures. The type of interaction energies calculated during this analysis shows that the steric contribution (the Lennard–Jones energy) is predominant in the most stable complexes. This behavior is confirmed by CoMFA models obtained through the three-dimensional QSAR. The steric model field (or at least a combination of the two models) produces a very consistent activity predictor model—suggesting the descriptors included in this field and the structural information obtained by molecular docking analysis—as a basis for designing new effective drugs. These predicted models and structural information confirm the molecule ranking reported in Table 1 and identify Bafetinib as the best candidate for a hiPDGFR ligand/inhibitor.

Although this analysis was carried out with bioinformatics tools and needs experimental evidence and confirmation, we can assert that it has the potential to create a platform for developing new drugs or repositioning already-known drugs. Though the predicted model is an excellent starting point for developing an ideal hiPDGFRα ligand/inhibitor, all 27 selected compounds have the potential to effectively bind and inhibit hiPDGFRα activity based on predicted geometries and affinities. This is partially supported by the fact that compounds such as Imatinib, Nilotinib, Crenolanib, and Nintedanib, which are included in this set, have already been experimentally shown to have activity against hiPDGFR [1,4,8,9,11,15]. Moreover, Bafetinib is already known to inhibit the Bcr/Abl fusion protein tyrosine kinase and the Src-family member Lyn tyrosine kinase for treating Bcr-Abl+ leukemias, including chronic myelogenous leukemia (CML) and Philadelphia+ acute lymphoblastic leukemia [33,34]. Further studies are necessary to determine whether these compounds should proceed to the pre-clinical and clinical phases of evaluation. As reported in a previous study on Nintedanib [35], it will be crucial to carefully adjust the dosage of these new drugs in a real-life setting to ensure their optimal efficacy and safety for patients. This requires close monitoring of potential side effects and close collaboration between healthcare providers and patients to make necessary adjustments based on individual patient factors, such as age, weight, and comorbidities.

## 4. Materials and Methods

### 4.1. Virtual Screening of the Small Compounds Database

The structure-based virtual screening of compounds, such as drugs, food constituents, and natural products, against the human intracellular PDGFRα was performed using the MTiOpenScreen web server [36,37] that is based on Autodock Vina-based docking genetic algorithm and analyses 5 compounds libraries: iPPI-lib, diverse-lib, Drugs-lib, Food-lib, and NP-lib. These drug-like chemical libraries contain modulators of protein–protein interactions, diverse chemical compounds, purchasable approved drugs, food constituents, and natural products, respectively, currently for a total of about 7200 molecules. After the preparation of the receptor (pdbID: 6jol) [12] removing Imatinib and water molecules, the PDB file was uploaded on the server setting the grid center coordinates (64.868, 43.016, 0.324) and the size of the search space (30 × 28 × 36 Å), keeping all the other parameters as default. 

### 4.2. Molecular Docking

Twenty-seven molecules were mainly selected on the basis of the best binding energies obtained from the results of the MTiOpenScreen web server. To check these results, we carried out a further docking analysis using the SwissDock web server, as previously reported [38], based on the EADock DSS algorithm [39,40], and setting the docking type as accurate without a definition of the region of interest to include the whole protein in the search space. The final geometry of the complexes obtained was rendered by PyMol software (The PyMOL Molecular Graphics System, Version 2.5.2, Schrödinger, LLC., Cambridge, MA, USA), and the 3D representation of the interaction was obtained by the Protein-Ligand Interaction Profiler (PLIP) web server [41].

### 4.3. Molecular Dynamics

The molecular dynamics simulation of the best 4 complexes obtained (including the Imatinib/receptor complex as reference) was performed using Gromacs software version 2022.4 [42], the CHARMM36 force field updated July 2022 [43], the SPC216 water model, and the CGenFF server for generating ligand topologies [43]. Each dynamics simulation started with solvation, neutralization adding three chlorine ions, and minimization of the system using the steepest descend algorithm for 50,000 steps. Next, the NVT and NPT equilibration stage was performed in a 100 ps run, stabilizing the temperature at 300 K (using V-rescale Berendsen thermostat) and the pressure at 1 bar (using C-rescale pressure coupling), respectively. Finally, MD simulation was run for 50 ns, and the rms tools of Gromacs were used to determine the root mean square deviation (RMSD) of each ligand trajectory in relation to the protein backbone and, also, to calculate the protein backbone RMSD relative to the energy minimized conformation. The Coulombic and Lennard–Jones interaction energies were calculated using the capability of Gromacs to decompose the short-range non-bonded energies.

### 4.4. Three-Dimensional Quantitative Structure–Activity Relationship (3D-QSAR) and ADMET Analyses

To simplify the design of selective human intracellular PDGFRα inhibitors, the three-dimensional quantitative structure–activity relationship (3D-QSAR) analysis was carried out using the comparative molecular fields analysis (CoMFA) [44] available on the web portal www.3d-qsar.com [45]. To achieve this analysis, we first performed an alignment of molecules’ conformations and a molecular interaction fields (MIFs) calculation of the 27 selected molecules. We set all parameters as default. The two CoMFA potentials obtained, called steric (STE) and electrostatic (ELE), were calculated by means of the Lennard–Jones and Coulomb law definitions, respectively.

The ADMET (absorption, distribution, metabolism, excretion, and toxicity) analysis was carried out using the screening section of ADMETlab 2.0 web server [46] and uploading the SMILES notation of the 27 selected molecules. In addition to the most common molecular descriptors, the results provide four acceptance indexes called Lipinski, Pfizer, GSK, and Golden Triangle. The Lipinski rule is related to absorption and permeability, while the Pfizer rule is related to toxicity. Compounds satisfying the GSK and Golden Triangle rule may have a more favorable ADMET profile.

## Figures and Tables

**Figure 1 ijms-24-09623-f001:**
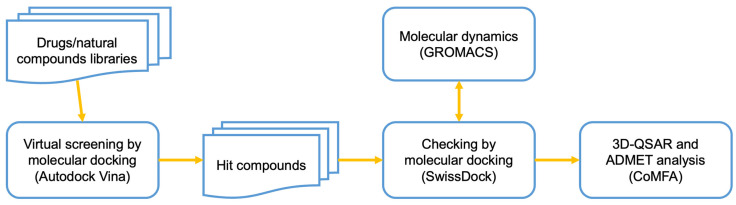
Schematic representation of the methodology and workflow implemented for the SBDD.

**Figure 2 ijms-24-09623-f002:**
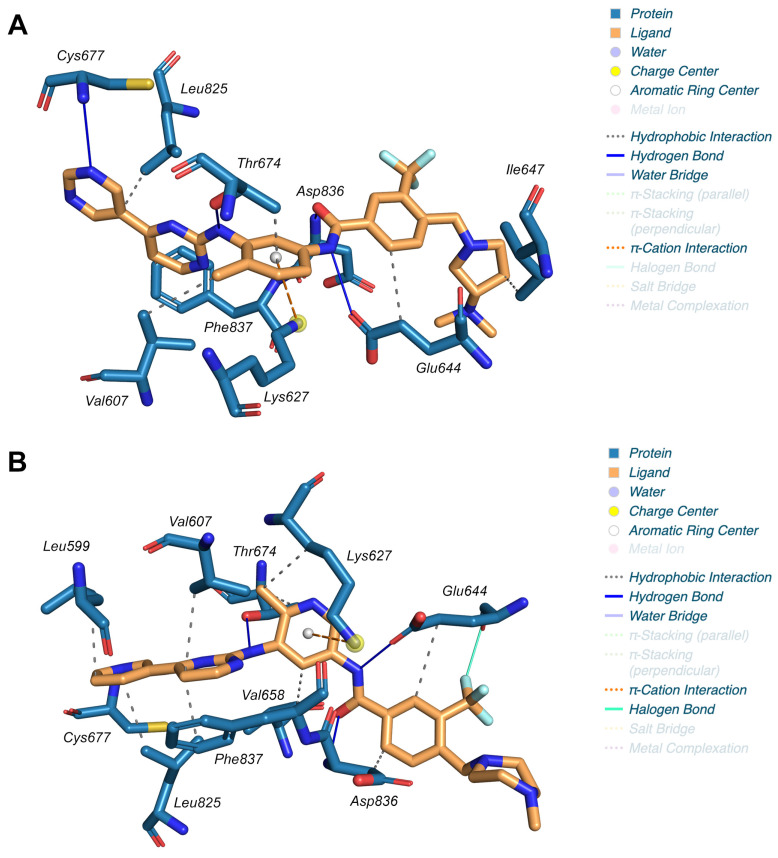
Complex geometries prediction by molecular docking analysis: amino acids and the ligand of (**A**) Bafetinib/hiPDGFRα and (**B**) Flumatinib/hiPDGFRα complex involved in the interaction are shown as sticks. Weak bonds are also reported and emphasized in the legends.

**Figure 3 ijms-24-09623-f003:**
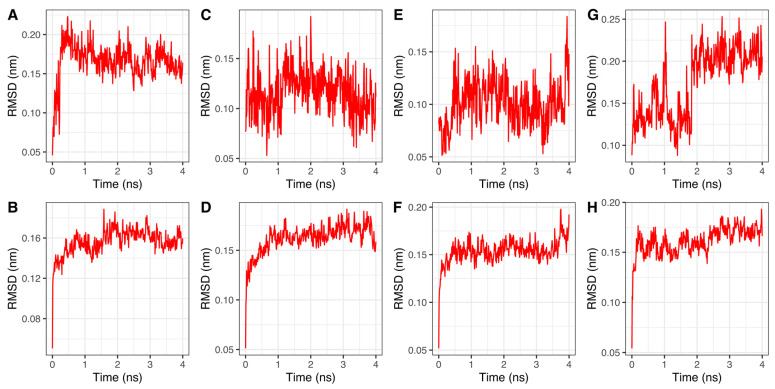
RMSDs of ligands and complexes trajectories: (**A**,**C**,**E**,**G**) RMSDs of Bafetinib, Radotinib, Imatinib, and Flumatinib versus protein backbone, respectively; (**B**,**D**,**F**,**H**) RMSDs of hiPDGFRα versus protein backbone complexed with Bafetinib, Radotinib, Imatinib, and Flumatinib, respectively.

**Figure 4 ijms-24-09623-f004:**
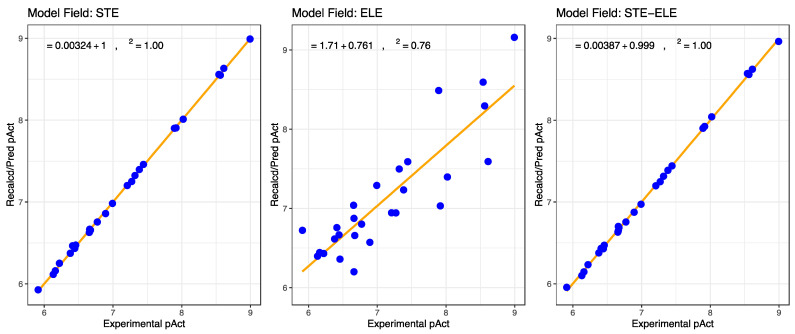
CoMFA model prediction values of activities (Recalcd/Pred pAct) versus experimental activities obtained by molecular docking analysis considering the optimal PC values.

**Table 1 ijms-24-09623-t001:** The best compounds selected after the structure-based virtual screening and their affinities for the hiPDGFRα obtained after the molecular docking analysis by the SwissDock web server.

Compound	Database	Δ*G* (kcal/mol)	*K_d,pred_* (M)
Bafetinib	Drugs-lib	−12.26	1.01 × 10^−9^
Radotinib	Drugs-lib	−11.74	2.43 × 10^−9^
Imatinib	Drugs-lib	−11.67	2.74 × 10^−9^
Flumatinib	Drugs-lib	−11.64	2.88 × 10^−9^
Nilotinib	Drugs-lib	−10.93	9.53 × 10^−9^
Ag-13958	Drugs-lib	−10.79	1.21 × 10^−8^
Tg100-801	Drugs-lib	−10.76	1.28 × 10^−8^
Ditercalinium	Drugs-lib	−10.15	3.60 × 10^−8^
Dasatinib	Drugs-lib	−10.06	4.14 × 10^−8^
R428_Bemcentinib	Drugs-lib	−9.98	4.78 × 10^−8^
Glisolamide	Drugs-lib	−9.91	5.35 × 10^−8^
MolPort-002-524-598	NP-lib	−9.83	6.19 × 10^−8^
Bms-833923	Drugs-lib	−9.53	1.01 × 10^−7^
Benfluorex	Drugs-lib	−9.40	1.28 × 10^−7^
MolPort-039-052-621	NP-lib	−9.23	1.69 × 10^−7^
MolPort-009-018-791_Curcumin	NP-lib	−9.10	2.12 × 10^−7^
47194043_iPPI	iPPI-lib	−9.08	2.18 × 10^−7^
Crenolanib	Drugs-lib	−9.08	2.18 × 10^−7^
MolPort-001-740-946_luteolin-7-O-glucuronide	NP-lib	−9.07	2.21 × 10^−7^
Nintedanib	Drugs-lib	−8.80	3.48 × 10^−7^
24301892_iPPI	iPPI-lib	−8.78	3.60 × 10^−7^
24824231_div	Diverse-lib	−8.74	3.88 × 10^−7^
MolPort-001-741-358_EGCG	NP-lib	−8.70	4.18 × 10^−7^
29215783_iPPI	iPPI-lib	−8.48	6.00 × 10^−7^
MolPort-001-740-557_Quercetin	NP-lib	−8.40	6.88 × 10^−7^
Hispaglabridin_B	Food-lib	−8.36	7.38 × 10^−7^
MolPort-001-768-161_Binaphthalene	NP-lib	−8.06	1.23 × 10^−6^

**Table 2 ijms-24-09623-t002:** Average short-range interaction energies calculated by energy module of Gromacs.

Complex	Coulombic Interaction Energy (kJ/mol)	Lennard–Jones Energy (kJ/mol)
Bafetinib/hiPDGFRα	−135.72 ± 0.96	−257.94 ± 0.60
Radotinib/hiPDGFRα	−142.66 ± 1.10	−234.45 ± 1.50
Imatinib/hiPDGFRα	−121.95 ± 1.20	−234.49 ± 1.80
Flumatinib/hiPDGFRα	−135.79 ± 2.10	−244.41 ± 3.00

**Table 3 ijms-24-09623-t003:** Summary of the best CoMFA models.

Field	*r* ^2^	*q* ^2^	Optimal PC
STE	1.000	0.525	5
ELE	0.761	0.574	4
STE-ELE	0.999	0.692	6

**Table 4 ijms-24-09623-t004:** ADMET results. Indexes and some descriptors are reported (part 1).

Compound	Lipinski	Pfizer	GSK	Golden Triangle	MW	LogP	LogD	nHA	nHD	TPSA
Recommended Range					100/600	0/3	1/3	0/12	0/7	0/140
Benfluorex	Accepted	Rejected	Rejected	Accepted	351.14	4.229	4.039	3	1	38.33
Glisolamide	Accepted	Accepted	Rejected	Accepted	434.16	2.953	1.325	9	3	130.4
Imatinib	Accepted	Accepted	Rejected	Accepted	493.26	3.805	3.144	8	2	89.51
Nintedanib	Accepted	Accepted	Rejected	Rejected	539.25	3.466	3.036	9	2	101.47
Nilotinib	Accepted	Accepted	Rejected	Rejected	529.18	4.894	3.881	8	2	100.85
Radotinib	Accepted	Accepted	Rejected	Rejected	530.18	4.574	3.643	9	2	113.74
Flumatinib	Accepted	Accepted	Rejected	Rejected	562.24	4.043	3.27	9	2	102.4
Ag-13958	Accepted	Accepted	Rejected	Accepted	467.19	4.625	3.476	8	3	100.52
R428_Bemcentinib	Accepted	Accepted	Rejected	Rejected	506.29	4.54	4.017	8	3	101.74
Bafetinib	Accepted	Accepted	Rejected	Rejected	576.26	4.276	3.518	9	2	102.4
Ditercalinium	Rejected	Rejected	Rejected	Rejected	718.4	7.562	5.014	8	2	64.28
Bms-833923	Accepted	Accepted	Rejected	Accepted	473.22	5.193	3.902	6	3	82.17
Tg100-801	Rejected	Accepted	Rejected	Rejected	579.2	6.745	4.857	8	1	92.7
Hispaglabridin_B	Accepted	Rejected	Rejected	Rejected	390.18	6.757	5.324	4	1	47.92
Crenolanib	Accepted	Accepted	Rejected	Accepted	443.23	4.34	3.093	7	2	78.43
Dasatinib	Accepted	Accepted	Rejected	Accepted	487.16	2.807	2.922	9	3	109.74
MolPort-001-740-557_Quercetin	Accepted	Accepted	Accepted	Accepted	302.04	2.155	1.767	7	5	131.36
MolPort-001-768-161_Binaphthalene	Accepted	Rejected	Rejected	Accepted	254.11	6.02	4.668	0	0	0
MolPort-001-741-358_EGCG	Rejected	Accepted	Rejected	Accepted	458.08	1.893	0.652	11	8	197.37
MolPort-009-018-791_Curcumin	Accepted	Accepted	Accepted	Accepted	368.13	2.742	2.82	6	2	93.06
MolPort-039-052-621	Rejected	Accepted	Rejected	Rejected	542.12	5.021	3.33	10	5	162.98
MolPort-002-524-598	Rejected	Accepted	Rejected	Rejected	596.14	−0.524	−0.3	16	10	269.43
MolPort-001-740-946_Luteolin-7-glucuronide	Rejected	Accepted	Rejected	Accepted	462.08	0.864	0.745	12	7	207.35
29215783_iPPI	Accepted	Rejected	Rejected	Accepted	410.08	5.88	4.167	4	0	48.67
24301892_iPPI	Accepted	Accepted	Rejected	Accepted	454.13	5.425	4.104	7	1	77.75
47194043_iPPI	Accepted	Rejected	Rejected	Accepted	354.14	4.824	3.525	4	2	61.96
24824231_div	Accepted	Accepted	Accepted	Accepted	330.1	2.863	2.829	5	2	75.27

Abbreviations: molecular weight (MW), logarithm of the n-octanol/water distribution coefficient (LogP), logarithm of the n-octanol/water distribution coefficients at pH = 7.4 (LogD), number of hydrogen bond acceptors (nHA), number of hydrogen bond donors (nHD), and topological polar surface area (TPSA).

**Table 5 ijms-24-09623-t005:** ADMET results. Indexes and some descriptors are reported (part 2).

Compound	QED	NP-Likeness	Caco-2	MDCK	Carcinogenicity	SkinSen	EC	EI	Respiratory	LD50 Oral
Recommended Range	>0.67	−5/5	>−5.15	>2.0 × 10^−6^	0/0.3 Excellent; 0.3/0.7 Medium; 0.7/1 Poor					0
Benfluorex	0.603	−0.95	−4.62	2.19 × 10^−5^	0.046	0.033	0.003	0.011	0.686	0
Glisolamide	0.613	−1.57	−5.447	1.56 × 10^−5^	0.026	0.109	0.003	0.007	0.007	0
Imatinib	0.424	−1.477	−5.576	7.27 × 10^−6^	0.044	0.946	0.003	0.007	0.992	0
Nintedanib	0.272	−1.176	−5.716	2 × 10^−5^	0.078	0.75	0.003	0.007	0.694	0
Nilotinib	0.302	−1.657	−5.112	1.12 × 10^−5^	0.027	0.932	0.003	0.009	0.928	0
Radotinib	0.325	−1.62	−5.065	1.06 × 10^−5^	0.021	0.944	0.003	0.009	0.954	0
Flumatinib	0.361	−1.566	−5.492	7.93 × 10^−6^	0.028	0.948	0.003	0.006	0.988	0
Ag-13958	0.314	−1.989	−5.46	6.03 × 10^−6^	0.591	0.245	0.003	0.013	0.986	0
R428_Bemcentinib	0.364	−0.866	−5.342	6.99 × 10^−6^	0.95	0.581	0.003	0.007	0.74	0
Bafetinib	0.327	−1.607	−5.658	5.79 × 10^−6^	0.036	0.949	0.003	0.006	0.95	0
Ditercalinium	0.149	0.003	−5.949	6.59 × 10^−6^	0.086	0.962	0.003	0.009	0.999	0
Bms-833923	0.294	−0.978	−5.526	1.54 × 10^−5^	0.169	0.392	0.003	0.01	0.896	0
Tg100-801	0.166	−1.068	−5.193	1.92 × 10^−5^	0.254	0.477	0.003	0.009	0.743	0
Hispaglabridin_B	0.7	2.088	−5.06	1.87 × 10^−5^	0.906	0.854	0.003	0.111	0.884	0
Crenolanib	0.504	−0.993	−5.421	9.76 × 10^−5^	0.825	0.543	0.003	0.008	0.81	0
Dasatinib	0.493	−1.724	−4.871	1.25 × 10^−5^	0.948	0.139	0.003	0.008	0.846	0
MolPort-001-740-557_Quercetin	0.434	1.701	−5.204	7.69 × 10^−5^	0.05	0.919	0.007	0.936	0.072	0
MolPort-001-768-161_Binaphthalene	0.411	−0.156	−4.691	1.45 × 10^−5^	0.861	0.945	0.011	0.983	0.019	0
MolPort-001-741-358_EGCG	0.212	1.65	−6.717	5.05 × 10^−6^	0.034	0.969	0.003	0.936	0.021	0
MolPort-009-018-791_Curcumin	0.548	0.722	−4.834	1.63 × 10^−5^	0.706	0.958	0.007	0.792	0.951	0
MolPort-039-052-621	0.225	0.947	−5.059	2.03 × 10^−5^	0.39	0.94	0.003	0.912	0.816	0
MolPort-002-524-598	0.137	1.984	−6.448	1.21 × 10^−5^	0.243	0.674	0.003	0.222	0.018	0
MolPort-001-740-946_Luteolin-7-glucuronide	0.253	1.771	−6.471	3.08 × 10^−5^	0.279	0.079	0.003	0.013	0.058	0
29215783_iPPI	0.388	−0.195	−4.916	3.44 × 10^−5^	0.575	0.033	0.003	0.513	0.704	0
24301892_iPPI	0.471	−1.923	−4.942	1.95 × 10^−5^	0.554	0.042	0.003	0.054	0.838	0
47194043_iPPI	0.499	−0.597	−5.034	7.22 × 10^−6^	0.465	0.882	0.003	0.623	0.684	0
24824231_div	0.725	−1.188	−5.289	1.36 × 10^−5^	0.427	0.854	0.003	0.036	0.017	0

Abbreviations: drug-likeness based on the concept of desirability (QED), Natural Product-likeness score (NP-likeness), human colon adenocarcinoma cell lines permeability (Caco-2), Madin−Darby Canine Kidney cells permeability (MDCK), skin sensitization (Skin Sen), eye irritation/corrosion (EI/EC) potential, respiratory toxicity (Respiratory), and acute toxicity during oral administration (LD50 oral).

## Data Availability

Not applicable.

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
