# Peer review of "Screening and Analysis of Possible Drugs Binding to PDGFRα: A Molecular Modeling Study"

_ijms, 2023, doi:10.3390/ijms24119623_

Round 1

Reviewer 1 Report

The authors performed large scale virtual screening with the human intracellular PDGFR against  7200 drugs and natural compounds. The drugs Bafetinib, Radotinib, Flumatinib and Imatinib showed the higher affinity for this tyrosine kinase receptor, while the natural products included in this group, such as curcumin, luteolin and EGCG. The study conducted is systematic and interesting. However I have few concerns before accepting this paper:

Before accepting this paper, I have few major concerns to be addressed:

Since this is a computational study, write the advantages of molecular modeling in introduction.

The paper is not well written and well referenced (Introduction section in particular). Important articles should be emphatically introduced (PMC9133930).

I also recommend the authors to perform molecular dynamics simulations for at least 50 nanoseconds for stability analysis accompanied with binding free energy analysis (Since the paper lack intensive computations).

To be honest, there are many articles about identifying PDGFR inhibitors. What is the highlight of this manuscript? Author should clarify the differences between this article and many previous similar articles about this topic.

It is difficult to grasp the overall direction of the paper. The authors should add a flowchart that schematically represents overall methodology adopted in the paper.

Reviewer 2 Report

This is a well-designed and beautiful molecular modeling study describing a screening analysis to identify possible drugs binding to PDGFRa. Although the manuscript is well written and provide an interesting information, this reviewer missed some biological information regarding the PDGFRa expression in different cancer subtypes. The major goal of the study is finding molecules that bind PDGFRa to treat cancers overexpressing this marker. Therefore, no information was provided. The discussion section is poorly informative. Thus, authors could increase discussion section provided some biological information and providing some association with cancers overexpressing PDGFRa.

Reviewer 3 Report

The manuscript submitted by Matteo Mozzicafreddo et al reported on the “Screening and analysis of possible drugs binding to PDGFR a molecular modeling study” In this work, drugs or natural compounds were screened and 27 potential candidates were analyzed to bind to platelet-derived growth factor receptor. Molecular Docking, Molecular Dynamics were used to analyze those materials.  

The paper was kind of straightforward and calculation experiments were performed. But the manuscript does not contain enough work. Besides, I don’t see the significance of doing this work as scanning the databases and analyzing existing data is very common in current science especially when the AI can almost do all these jobs. 

First, as a molecule-related paper, the paper only contains 29 references which is unbelievable. It lacks many proper citations.  

Second, when finishing reading the calculation part, the readers will definitely look for your real test. Which molecular candidates perform the best and does it match the results of your calculation and database? But it directly comes to an end, which can show nothing of the significance of this work. 

Please figure out or address the importance of your work. It is pretty common in drug development that good candidates fail to perform well due to diverse reasons. Only data search and calculation can’t be turned into a good manuscript. It is not a chemistry mechanism study.  

I believe the authors need to think more about what you are studying. I will suggest major revision.

Round 2

Reviewer 1 Report

No further comments

Author Response

Many thanks for all your helpful previously comments.